



**Frozen debris lobe morphology and movement: an**
**overview of eight dynamic features, southern Brooks**
**Range, Alaska**
**Margaret M. Darrow,[1] Nora L. Gyswyt[1], Jocelyn M. Simpson[1], Ronald P.**
**Daanen[2], Trent D. Hubbard[2]**
[1]Department of Mining and Geological Engineering, University of Alaska Fairbanks,
Fairbanks, Alaska 99775, USA
[2]Alaska Division of Geological & Geophysical Surveys, Fairbanks, Alaska 99709, USA
*Correspondence to*: Margaret M. Darrow (mmdarrow@alaska.edu)
**Abstract**
Frozen debris lobes (FDLs) are elongated, lobate permafrost features, many of which are
present within the Brooks Range of Alaska. We present a comprehensive overview of eight
FDLs within the Dalton Highway corridor, including their catchment geology and rock
strengths, lobe soil characteristics, surface movement measurements collected between 2012
and 2015, and analysis of historic and modern imagery from 1955 to 2014. Field mapping
and rock strength data indicate that the metasedimentary and metavolcanic bedrock forming
the majority of the lobe catchments has very low to medium strength and is heavily fractured,
thus easily contributing to FDL formation. The eight investigated FDLs consist of platy rocks
typical of their catchments, organic debris, and an ice-poor soil matrix; massive ice, however,
is present within FDLs as infiltration ice, concentrated within cracks open to the surface.
Exposure of infiltration ice in retrogressive thaw slumps (RTSs) and associated debris flows
lead to increased movement and various stages of destabilization, resulting in morphological
differences among the lobes. Analysis of historic imagery indicates that movement of the
eight investigated FDLs has been asynchronous over the study period. Since 1955, six of the
eight investigated lobes demonstrated an increase in movement rates. The formation of
surface features, such as cracks, scarps, and RTSs, suggest that the increased movement rates
correlate to general instability, and even at their current distances, FDLs are impacting





infrastructure through increased sediment mobilization. FDL-A is the largest of the
investigated FDLs. As of August 2015, FDL-A was 39.2m from the toe of the Dalton
Highway embankment. Based on its current distance and rate of movement, we predict that
FDL-A will reach the current Dalton Highway alignment by 2023.
**1    Introduction**
A warming climate has been identified as unequivocal by the Intergovernmental Panel on
Climate Change (IPCC), with greater and faster temperature increase and an overall
precipitation increase demonstrated at northern latitudes (Stocker et al., 2013). Analysis of
field data collected throughout Arctic and sub-Arctic areas corroborates with IPCC's findings,
demonstrating an overall permafrost temperature rise (Christiansen et al., 2010; Romanovsky
et al., 2010; Smith et al., 2010). Slopes in permafrost areas are in danger of instability with
rising temperatures, as the ice within the soil on the slopes melts and causes loss of soil
strength (Geertsema et al., 2006; Gude and Barsch, 2005; Harris et al., 2008a, 2008b;
Lewkowicz and Harris, 2005; Swanger and Marchant, 2007). Slope instability presents a risk
to adjacent infrastructure especially where roads and utilities pass through mountainous
regions. An increase in infrastructure construction may occur in northern regions, including
Alaska, as Arctic countries focus on economic development (EOP, 2014; Sevunts, 2013).
Thus, recognizing areas of slope instability and quantifying historic and potential movement
become progressively important as climate changes and northern regions see increasing
development.
An example of previous development in the Alaskan Arctic was the construction of the Trans
Alaska Pipeline System (TAPS) and supporting infrastructure, including the Dalton Highway,
which opened a corridor within the Brooks Range. In the late 1970's and early 1980's, those
mapping the geology and geologic hazards along the Dalton Highway corridor noted the
presence of elongated, lobate features along slopes adjacent to the highway, thought to be
inactive at that time (Hamilton, 1978, 1979, 1981; Kreig and Reger, 1982; Brown and Kreig,
1983). These features were "rediscovered" in 2008, partially due to the fact that they were
indeed actively moving. When originally mapped, these features were identified as flow
slides or rock glaciers; however, ongoing investigations indicate that they are different from
rock glaciers in their source, composition, rate and mechanism of movement, and vegetation
coverage (Daanen et al., 2012; Simpson et al., in press). Because of these differences, these





permafrost features were given the new name *frozen debris lobes* (FDLs) (Daanen et al.,
2012). As the resolution of freely-available satellite imagery improves, we continue to
identify additional FDLs, with nearly 160 FDLs located thus far within the Brooks Range
(Figure S1).
Much work has been done to describe and categorize mass wasting features on permafrost-
stabilized slopes in mountain regions (French, 2007; Gorbunov and Seversky, 1999; Gruber
and Haeberli, 2007; Humlum, 1998a; Kääb et al., 2007; Matsuoka et al., 2005; Wahrhaftig
and Cox, 1959). FDLs are the latest features to be defined (Daanen et al., 2012), representing
one part in the continuum of permafrost creep features (Haeberli et al., 2006). Since FDLs
were referred to previously as rock glaciers, we include a brief summary of these features for
comparison. Rock glaciers are described in many cold climate regions (Ballantyne et al.,
2009; Barsch, 1977; Berthling et al., 2003; Bollman et al., 2015; Brenning and Azocar, 2010;
Calkin et al., 1998; Farbrot et al., 2007; Haeberli et al., 2006; Humlum, 1998a, 1998b; Isaksen
et al., 2000; Kääb et al., 1997, Wahrhaftig and Cox, 1959; Wirz et al., 2015), typically
forming on talus slopes, at the base of cliffs, or within cirques (Davis, 2001; Embleton and
King, 1975). These features often demonstrate surfaces composed of blocky debris, which
may be underlain by finer material (Embleton and King, 1975; Ikeda and Matsuoka, 2006;
Wahrhaftig and Cox, 1959). In Alaska's Brooks Range, Calkin et al. (1987) hypothesized that
rock glaciers formed just after Pleistocene glacial retreat. Ellis and Calkin (1984) suggested
that these rock glaciers probably formed from increased rockfall from oversteepened valleys
and cirque walls after the late Pleistocene glaciation, resulting in mostly glacier-cored rock
glaciers.
In terms of lobe geometry, Matsuoka et al. (2005) describe a classification for rock glaciers
and solifluction lobes. FDLs most resemble these authors' description for "pebbly" rock
glaciers; however, the dimensions of FDLs are typically much greater and their debris does
not resemble pebbles. Rock glaciers typically move 1m $yr^{-1}$ or less, although recent
measurements show rates as high as 6.3 m $yr^{-1}$ (Micheletti et al., 2015; Wirz et al., 2015). We
show in this paper that FDLs move at rates an order of magnitude greater than those typical of
rock glaciers. Another characteristic of FDLs that sets them apart from most rock glaciers is
their vegetative cover. Many of the FDLs within the Dalton Highway corridor support mature
white spruce forests on their surfaces and all support brushy vegetation; the trees indicate
movement of the underlying FDL by their orientations away from vertical. Forested rock
glaciers do exist, such as Tien Shan rock glaciers in Central Asia (Sorg et al., 2015) and the



Slims River lobate rock glacier in Canada (Blumstengel and Harris, 1988); however, their
slower rates of movement and, in the case of the Slims River rock glacier, its icy matrix set
these rock glaciers apart from FDLs, as we show below.
Field investigations of FDLs began in 2008 with preliminary differential global positioning
system (DGPS) measurements, soil pits, and field observations on FDL-A, with some work on
nearby FDL-B, -C, and -D (Daanen et al., 2012). Due to its close proximity to the Dalton
Highway, we conducted a subsurface investigation of FDL-A jointly with the Alaska
Department of Transportation and Public Facilities (ADOT&PF) in September 2012 (Darrow
et al., 2012, 2013). Simpson et al. (in press) focused on the geotechnical and GIS analysis of
FDL-A, presenting the results of strength testing of frozen soil samples and a slope stability
analysis, and the initial results of a GIS protocol by which to examine FDLs. Results from
these early investigations indicate that FDLs are mainly composed of a fine-grained soil
matrix, but also contain rocks and organic debris. Data from instrumentation installed within
FDL-A indicate that this frozen debris lobe moves through a combination of shear in a zone
20.6 to 22.8m below ground surface (bgs) and through temperature-dependent internal flow
(Darrow et al., 2015; Simpson et al., in press). The average internal temperature of the lobe is
-1.1°C below the depth of zero annual amplitude, whereas the temperature of the adjacent
permafrost is -2.2°C. This summary represents the bulk of work conducted thus far on FDLs,
ranging from setting the stage for FDL analysis through the presentation of preliminary results
and interpretation (Daanen et al., 2012), to a more in-depth focus on FDL-A (Darrow et al.,
2012, 2013, 2015; Simpson et al., in press).
In 2013 we expanded the Area of Interest (AOI) to include eight FDLs (Figure 1). Since
expanding the investigation, we have traveled to the field two to three times a year to collect
DGPS measurements of the FDL surfaces, as well as samples of soil, rock, water, and ice.
Our field investigations and observations led us to the following questions. 1) How does the
bedrock source geology contribute to FDL morphology? 2) Are the investigated FDLs
consistent in composition and morphology? 3) Has the movement of these FDLs been
synchronous? 4) Have their rates of movement changed over time? 5) How can we describe
the origin of these features? 6) How are FDLs impacting infrastructure? In an effort to
answer these questions, in this paper we present for the first time a comprehensive overview
of eight different FDLs within the Dalton Highway corridor. Within this overview, we detail
catchment geology, and measured rock strengths and lobe soil characteristics. To illustrate
current and historic rates of movement, we present DGPS surface movement measurements



collected between 2012 and 2015; and analysis of aerial imagery, satellite imagery, and Light
Detection and Ranging (LiDAR) data collected between 1955 and 2014. We also present the
results of stable isotope analysis of ice and water samples and radiocarbon dating results of
organic soil samples from FDL-A, used to determine its origin and long-term record of
movement.
## 2  Methods
### 2.1  Fieldwork and sample collection
We selected eight FDLs to assess their geohazard potential based on their size, evidence of
movement, and proximity to the Dalton Highway. The planned field study consisted of
measuring surface movement rates, and collecting samples for rock strength and soil index
property testing. During field visits, we also walked each of the FDLs to evaluate the lobes'
surface characteristics, identifying notable features such as scarps, retrogressive thaw slumps
(RTSs) and exposed ice, thermokarsts, split trees, etc. The field investigations sometimes
yielded unforeseen opportunities to collect samples for special testing, as discussed below.
To determine current rates and spatial variability of surface movement, we installed surface
marker pins on the eight FDLs. We began by installing the surface marker pins on FDL-A in
late October 2012, which we have since measured two to three times a year (typically in
March or April, June, and August). In June 2013, we installed surface marker pins on the
other seven FDLs under investigation. On each lobe, the surface marker pins were positioned
along a longitudinal profile from the catchment to the lobe toe, and along at least one cross-
sectional transect. We made repeated measurements of all surface marker pins in August
2013, June and August 2014, and May and August 2015. Measurements were made with a
DGPS unit, having horizontal and vertical accuracies of ± 5 cm. In addition to establishing
the surface marker pin locations, we located and mapped easily visible scarps in the less-
vegetated catchment areas with a hand-held GPS. RTSs are present in several FDL
catchments and on the lobes. On return visits, we remapped the head scarps of the RTSs to
determine rates of regression.
We collected rock and soil samples (Figure S2) to determine rock strengths and soil
engineering index properties. We sampled rocks from the catchment areas for strength
testing, while also updating the pre-existing geologic map of the area (Spangler and Hubbard,





in review). Once back in the laboratory, tests were made with a hydraulic point-load testing
device to determine the rocks' point load strength indices, which were converted to uniaxial
compressive strengths. On each lobe, we dug two 1m deep soil pits, examining the near
surface soils and collecting samples for standard engineering index property testing, including
moisture content [ASTM D2116], organic content [modified from AASHTO T267], grain size
distribution [AASHTO T27/T11, ASTM D422], and plasticity [ASTM D4318] (AASHTO,
2009; ASTM, 1990, 1998, 2000). All soil classifications are based on the Unified Soil
Classification System (USCS). We also sampled buried organic material from the lower
catchment of FDL-A, sending the sampled soil to Beta Analytic, Inc. for radiocarbon dating.
This laboratory service calibrated the results using databases associated with the 2013
INTCAL program (Reimer et al., 2013), and the resulting two-sigma calendar calibration
range is presented here. With this data, we hoped to determine when FDL-A started to move
out of the catchment, extending its record of movement beyond the available historic data.
The summer of 2014 was the wettest on record for parts of Interior Alaska. As a result,
rainfall exposed massive ice in several RTSs on FDL-7, -A, and -D. Seizing the opportunity
to determine the origin of the massive ice, in August 2014 we collected samples of exposed
ice on FDL-A in the lower RTS near the left flank (see lower arrow in Figure 2a), as well as
water samples from the creek that drains FDL-A and a puddle adjacent to the lobe during a
major rain event; in March 2015, we collected two samples of snow from the lobe. These
samples were submitted for analysis to the Alaska Stable Isotope Facility at the University of
Alaska Fairbanks' Water & Environmental Research Center. The purpose of the isotope
analysis was to determine the relative age of the ice, and thus identify its probable origin.
Stable isotope data were obtained using continuous-flow isotope ratio mass spectrometry
(CFIRMS). The $\delta 2H$ and $\delta 18O$ values were measured using pyrolysis-EA-IRMS. This
method utilized a ThermoScientific high temperature elemental analyzer (TC/EA) and Conflo
IV interface with a DeltaVPlus Mass Spectrometer. Stable isotope ratios were reported in $\delta$
notation as parts per thousand (‰) deviation from the international standards, V-SMOW
(Standard Mean Ocean Water). Typically, instrument precision is <3.0‰ for hydrogen and
<0.5‰ for oxygen.
**2.2 Historic image collection and analysis**
We acquired aerial and satellite imagery for the AOI from years between 1955 and 2014
(Table 1); images for each dataset were compiled into mosaics using the Agisoft Photoscan



and ENVI software. The mosaics were ortho-rectified according to the American Society of
Photogrammetry and Remote Sensing's (ASPRS) horizontal accuracy standards (ASPRS,
2015). In a GIS environment, we used contour lines derived from digital elevation models
(DEMs) produced from 2011 LiDAR and 2001 Interferometric Synthetic Aperture Radar
(IfSAR) data (1m and 5m resolution, respectively), GPS measurements, and field observations
as references for determining catchment and 2011 lobe extents. Next, we determined the
extent of each lobe for each year of available imagery (spatial limitations in imagery coverage
are summarized in Table 1). Because the FDLs demonstrate downslope movement with only
minor lateral spreading, longitudinal profile polylines oriented along the center of each lobe
served as the consistent reference from which to measure changes. Historic movement of
each FDL was analyzed by measuring the progression of the toe of the lobe between each set
of data years. Although not part of the rate analysis, we also include assessment of 2001 and
2002 Google Earth images of the AOI in the discussion.

## 3   Results

The results from the various FDLs have been grouped as appropriate in the following
discussion. When multiple FDLs are discussed or presented in figures and tables, they are
presented from the north to the south (see Figure 1 for FDL spatial distribution within the
AOI).

### 3.1   Catchments and rock data

Frozen debris lobes typically originate from catchments (Figure 2a), many of which may have
supported small cirque glaciers during early to mid-Pleistocene glacial advances in the area
(Hamilton, 1986). In some cases, FDLs have formed at the base of a slope rather than a
catchment, from the accumulation of loose colluvium (e.g., FDL-C), or from landslide
deposits (Figure 2b). The catchments of the eight FDLs presented here range from 121,000
$m^2$ (FDL-B) to 801,000 $m^2$ (FDL-A), with an average size of 414,000 $m^2$ (Table S1).
The catchments of the investigated FDLs range from bowl-like and well-defined (FDL-11, -B,
-A, -D, -5), to flatter with more open slopes (FDL-7, -4). The upper portions of the
catchments typically consist of exposed rock talus and solifluction lobes supporting shrubby
vegetation (Figure 2c). The major sources of debris coming into the catchments are rock fall
and solifluction (Daanen et al., 2012; Spangler et al., 2013).





The bedrock contributing to the studied catchment areas consists of heavily fractured,
metasedimentary and metavolcanic rocks, including phyllite, slate, metasiltstone,
metasandstone, greywacke, and conglomerate, with minor amounts of limestone, marble, and
igneous intrusions (Figure S2; see Table S2 for rock unit descriptions). The joint spacing is
typically less than 30mm. The rocks tested had strengths ranging from 14.0 to 77.3 MPa
(Figure S2, Table S1), which covers the range from very low strength to medium strength
(Kehew, 2006). It should be noted that testing was conducted on samples that were
competent enough to be collected and transported from the field, which suggests that these
strength values are an overestimate of the actual rock strengths in the field area. While the
bedrock in each catchment consists of different units, the commonality among all catchments
is the predominance of heavily fractured, platy, foliated rocks. Additionally, while some
samples demonstrated medium strength values, the test results and associated bedrock
geology indicate that most of the catchment areas are underlain by very low to low strength
rocks. The combination of low strength and high degree of fracturing suggests that much of
the bedrock can be treated as dense coarse soil (Milligan et al., 2005), thus easily contributing
to the formation of mass movement features such as FDLs.

## 3.2   Frozen debris lobe composition and morphology

FDLs are elongate lobate features. The areas of the eight FDLs presented here range from
83,000 m$^2$ (FDL-11) to 286,000 m$^2$ (FDL-A), with an average area of 149,000 m$^2$. Their
length-to-width ratios typically range from 4:1 to 7:1 (Table S3). An exception is FDL-C,
with a length-to-width ratio of 2:1. The rounder appearance of this lobe is most likely due to
its origin at the base of a slope rather than in a catchment, which limits both its supply of
debris and water. Most notable about the surface of FDL-C are the smaller, superimposed
surface lobes that form as the mass moves downslope (Figure 2d). These features are present
on several other lobes, including FDL-A. Analysis of subsurface data suggests that the
surface lobes form as faster internal flow within the active layer becomes sandwiched between
the cooling surface and the rising permafrost table in the fall (Darrow et al., 2015). Soil pits
excavated into the top 1 m of several of the lobes contained buried organic layers, possibly
buried as the surface was overrun by uphill material in a surface lobe.
Unlike their morphological cousins the rock glaciers, FDLs are composed of platy rocks
typical of their catchments, organic debris such as trees and shrubs, and a soil matrix typically
composed of silty sand with varying amounts of gravel, all of which is frozen (Table S3).



Where sampled, the upper 1m of tested FDL soils were moist to wet and slightly organic to
organic. Similar tests were conducted on the subsurface samples obtained from the 2012
drilling on FDL-A. Samples were collected from depths ranging from 2.9 to 24.8m bgs, and
tested uniformly as wet (when thawed) silty sand with gravel (Table S3), indicating a
consistency in the soil gradation and moisture content with depth that may occur for all FDLs.
Boreholes from the 2012 subsurface investigation intercepted no massive ice, and all samples
obtained from the drilling were ice-poor (i.e., <1% ice by volume).
While the soils are generally ice-poor, massive ice does exist within FDL-A and other FDLs.
Over several years, we have measured and observed the changes of two RTSs on the lobe's
surface (Figure 2a). For example, we photographed the change in the upper RTS on FDL-A
(Figure 3), which retreated up to 20m between 2011 and 2015. This rapid retreat is due
mostly to the melting of massive ice, which the significant rainfall of 2014 helped to expose.
While there are several different origins of massive ice in periglacial regions (Davis, 2001;
Washburn, 1985; Williams and Smith, 1989), we propose that the ice exposed in the FDLs is
infiltration ice, which forms as rain and snow melt quickly freeze after entering into cracks in
the ground; Tarussov (1992) used the term "infiltration ice" to describe a similar phenomenon
produced as summer melt infiltrates glacial ice. The ice we observed in the head scarps of the
RTSs was clear consisting of large crystals, and containing bubbles and strands of fungus
(Figure 4a inset). Observations of the head scarp in the lower RTS further support the
infiltration ice origin. In Figure 4a, the exposed massive ice is indicated in the center of the
photograph, with a buried organic layer vertically offset to its right and left, indicating
downslope movement. The location of the massive ice corresponds with an open surface
crack.
Figure 4b is a presentation of the isotope analysis results, with the Global Meteoric Water
Line (GMWL) plotted for comparison. The infiltration ice sample plots close to the average
of the creek and puddle samples collected during the 2014 summer. The average of the snow
samples is somewhat lighter. Isotope values from massive ice bodies taken from the literature
also are presented in Figure 4b. These include values from Pleistocene wedge ice near
Fairbanks, Alaska (Douglas et al., 2011), lateglacial and Holocene wedge ice near Barrow,
Alaska (Meyer et al., 2010), and a suite of wedge ice samples ranging in age from Pleistocene
to recent from northern Siberia (Meyer et al., 2002). This collection of data indicates that the
oldest ice has the lightest isotopic composition, which becomes heavier with decreasing age.





Most notable is that the infiltration ice sample from FDL-A is bracketed by recent and
subrecent wedge ice. The heavy isotopic composition of the infiltration ice and its similarity
to the creek and puddle samples supports the hypothesis that infiltration ice forms
predominantly from rain water entering cracks open at the surface.
Most of the investigated lobes support mature spruce forests whose leaning and cracked trees
alert the observer to subsurface movement. This is the case for three lobes, in particular, that
represent possible stages in FDL destabilization. Since beginning our field observations, we
have noticed increasing signs of instability in FDL-5. Its surface appears to be "deflating" as
evident by trees leaning towards its center on both the right and left flanks (Figure 2e). This
redistribution of its mass has resulted in oversteepening of the toe, measured at ~44º in 2015.
FDL-7 on the west side of the Dietrich River represents the next stage in destabilization. This
lobe also has deflated with trees leaning towards the center line of the lobe; however, at this
more advanced stage, the center has surged forward, forming a lower tongue shape (Figure
5a). The lower tongue is actively and quickly changing, with large exposures of bare mineral
soil and highly damaged spruce trees (Figure 5b). On the flanks where the lower tongue
begins, massive ice and ice-rich soil are exposed in RTSs that generate debris flows and
provide another source of surface water during the summer months (Figure 5c). Between
May and August 2015, significant erosion occurred, merging the head scarps of the RTSs near
the right and left flanks of FDL-7 into one RTS that spans the entire width of the lower
tongue.
The most advanced stage of destabilization is FDL-D. Between 1993 and 2001, a RTS
formed in FDL-D's lower catchment area. By 2010, transverse cracks in the catchment and
longitudinal cracks along the levees were visible and persistent throughout the winter,
indicating that the lobe was moving significantly throughout the year (see Figure 6a for an
example of transverse cracks within a winter aufeis deposit). Following the formation of the
RTS, FDL-D rapidly moved 316m downhill between 2002 and 2014. Although the
northernmost scarp has not changed significantly since 2011, other active RTSs continue to
enlarge. Our mapping of two other RTSs indicated up to 38m of head scarp retreat between
2011 and 2015. These head scarps expose massive ice, which melts and contributes to debris
flows. The debris flows cover much of the upper lobe area (Figure 6b), provide additional
sediment and water to the lobe, and increase the surface temperature of this already unstable
permafrost feature.



Downhill of the catchment, the surface of FDL-D is a jumbled landscape, full of cracks,
scarps, ponds, bare mineral soil, and crisscrossed vegetation that once was a mature spruce
forest. Figures 6c and 6d are two examples of the landscape and extreme movement of the
surface. In each photograph, a spruce tree is upside down with its roots (Figure 6c) or trunk
(Figure 6d) exposed, while the rest of the tree is completely consumed by the lobe.

### 3.3   Frozen debris lobe movement rates

Figures 7 and 8 are vector maps, illustrating the amount of movement measured on the lobe
surfaces between June 2013 and August 2015, as well as RTS head scarp retreat. The lobes
were divided into those demonstrating between 6 and 45m of movement (Figure 7), and those
demonstrating less than 6m of movement (Figure 8) during the measurement period. The
base map data for each of these images is 2011 LiDAR; as some of the FDLs have
demonstrated significant movement since then, the 2014 extent of each lobe is included.
Movement is generally downhill and parallel to each FDL's longitudinal profile (with FDL-7
moving generally eastward, and all other investigated FDLs moving generally westward).
Levees that formed along the lobe flanks demonstrate a component of movement away from
the center line, indicating some spreading of the lobe along its periphery (see FDL-7, -A, -C,
and -4 as examples). Additionally, the levees typically move slower than the rest of the lobe.
We observed a notable example of this differential movement in August 2014 on FDL-7 when
a recent debris flow along the levee margin was sheared forming echelon cracks within the
young deposit (Figure 5c). The average rates of movement for all FDLs for the 2013-2014
and 2014-2015 measurement periods are presented in Table S3. These values exclude
measurements taken on levees or above the lobes in their catchments. The 2014-2015 rates
range from 0.2m yr$^{-1}$ for FDL-11 to 13.3m yr$^{-1}$ for FDL-D, with FDL-A falling in between at
5.2m yr$^{-1}$.

### 3.4   Analysis of historic imagery

Figure 9 is a presentation of the change in extent of the eight investigated FDLs from 1955 to
2014. The distance between each pair of toe locations measured along the longitudinal center
line for two given data sets was divided by the time interval between data years, resulting in
an average movement rate for the time interval. Each rate was assigned to the latter of the two
data years (Table S4). These rates were plotted versus time, showing change in movement
rate over the total time period from 1955 to 2014 (Figure 10). A drawback to this approach is





that it does not capture uneven advancement of the toe due to topographic effects beneath the
lobe or differential movement within the lobe; however, the results allowed us to divide the
eight FDLs into two general groups, those with increasing rate trends (either steadily or
rapidly increasing; Figures 10a and 10b, respectively), and those with decreasing rate trends
(Figure 10c).  Only two of the eight FDLs have decreasing rate trends.
Analysis of the visual progression and rates indicates that movement of these FDLs has been
asynchronous over the study period.  For example, FDL-11 advanced nearly 10m yr$^{-1}$ in the
1970s, faster than any of the other FDLs at that time; however, our surface marker
measurements indicate that FDL-11 is currently moving only 0.2m yr$^{-1}$.  In contrast, FDL-D
experienced a rapid increase in movement in recent years, moving an average 32.1m yr$^{-1}$
between 2009 and 2011, with FDL-7 and FDL-5 demonstrating the next largest increases in
movement rates.  FDL-A on the other hand, has demonstrated a steady increase in its
movement rate since 1955, fitting a linear trend with a coefficient of correlation ($R^2$) of 0.88
over this period.
**4   Discussion**
The Brooks Range supported massive valley glaciers and ice caps several times during the
late Tertiary and Pleistocene (Hamilton 1986).  The catchments within the AOI supported
cirque glaciers during the Itkillik I advance (110-60 ka), but subsequent advances were not as
extensive (Hamilton 1978, 1986).  With the retreating ice, debris accumulated in the
catchment bottoms.  The platy and weak rocks typical of the area weathered to form the silty
sand with gravel soil matrix.  As the AOI was propitious for the formation of permafrost, the
debris froze as it accumulated, providing rheological properties that both countered erosional
processes and allowed flow.  Accumulation continued until the debris reached a "critical
mass" and began to flow out of the catchment areas.  The recharge of the debris in the
catchment areas is at a much slower rate than the movement rates of the individual lobes; thus,
this is the first and only journey these specific features will make downslope.  As indicated by
Daanen et al. (2012), the end of this mass movement process is an alluvial fan that forms
when an FDL completely destabilizes or when it reaches the river in the valley bottom, which
removes the toe.
When did these FDLs begin to flow from their catchments?  We focus on FDL-A for this part
of the discussion.  FDL-A is farther downslope than any of the other lobes, which suggests





that it either began to flow out of its catchment earlier or experienced rapid downslope
movement. From field observations and subsequent analysis of LiDAR data, we identified
the presence of relatively flat benches along the catchment slopes on either side of FDL-A.
We hypothesize that the benches represent the paleosurface of FDL-A before its downslope
movement began. Recreating the lobe within the catchment at this level provides a rough
volume estimate of 1,450,000 m$^3$ (Figure 11). To test our hypothesis and to determine when
the lobe left the catchment, in August 2015 we dug a trench in the active layer on the south
bench (see arrow and green dot in Figure 7b). The trench exposed the transition of colluvium
(brown organic silt) into typical FDL-A lobe soil (gray silty sand with gravel). Within the
lobe soil, we intercepted a buried organic layer, which we sampled for radiocarbon dating.
Based on the stratigraphy and its location on the bench, the organic layer was buried (possibly
by debris flow deposits) as the lobe surface was actively building. After the lobe reached its
critical mass, the center of FDL-A deflated (as currently observed on FDL-7 and FDL-5) as
the bulk of the lobe moved downslope, leaving behind the benches and the buried organic
material at the lobe's original elevation. The radiocarbon dating returned a calibrated result
(95% probability) of Cal AD 1220 to 1285 for the time of burial; thus the major downslope
movement of FDL-A began around 730 to 795 years ago.
Downslope movement of an FDL causes tension and shearing, which results in the formation
of surface cracks. All of the investigated FDLs support numerous transverse and longitudinal
cracks, and we suspect that these cracks contain infiltration ice. As a crack opens at the
surface due to movement, water entering the crack freezes forming infiltration ice; the crack
cannot close again, nor fill with debris. Thus, infiltration ice contributes to FDL movement
by providing additional lobe volume. An open, unfilled tension crack represents a break in
the subsurface lateral stress distribution (Cornforth, 2005); however, if filled with ice, stresses
developed in the upper lobe can be transmitted to the lower reaches. Finally, the ice volume
must be considered with increasing temperatures. Increased melting of infiltration ice will
lead to reduced soil strength and increased pore water pressure within the lobe (Simpson et al.,
in press), which will accelerate FDL movement. Based on the number of surface cracks, an
appreciable volume of the FDLs may be ice, certainly more than originally estimated from the
2012 subsurface investigation (Darrow et al., 2013).
All of the investigated FDLs have scarps or RTSs in their upper reaches (Figures 7 and 8).
Analysis of the historic images and Google Earth satellite imagery indicates that the change in
lobe morphology and formation of scarps on FDL-7 occurred between 1979 and 1993. The





scarps on FDL-A, -C, and -D formed later between 1993 and 2002; the scarps on the other
lobes are smaller and difficult to discern in the historic imagery.  In the case of FDL-D, the
scarp evolved into an active RTS, and subsequently the lobe moved rapidly downslope.  We
hypothesize that RTS formation is a key step in FDL destabilization.  The initial exposure of
bare mineral soil increases the surface temperature, which causes infiltration ice to melt
(Burn, 2000; Kokelj et al., 2009; Malone et al., 2013).  The meltwater forms debris flows that
cover a larger area of the lobe, further increasing the surface temperature and repeating the
cycle (Gooseff et al., 2009).  The debris flows also load the lobe surface with additional
sediment, potentially providing the extra driving force needed to initiate downslope
movement and the formation of transverse cracks.  The meltwater can infiltrate through the
now open cracks to the basal shear zone, increasing pore water pressure and further
accelerating the lobe's movement.  More movement perpetuates this process, resulting in
overall destabilization.
The underlying topography, in addition to RTS formation and increased surface temperatures,
may contribute to the destabilization of FDLs.  Examination of the topographic maps
generated from the 1955 imagery and the other historic images indicates that the drainages
downslope of FDL-11, -7, -D, and -5 have topographic constrictions that at some point
impeded downslope movement of these lobes.  The topographic constriction is most obvious
for FDL-7.  Sometime between 1979 and 1993, the lobe met this topographic narrowing,
which halted the movement of the toe of the lobe; however, by 1993 a small portion of the
lobe continued to flow forward, forming the lower tongue.  The subsequent shearing along the
flanks exposed infiltration ice, leading to growth of RTSs and acceleration of FDL-7's lower
tongue.  We suspect that FDL-5 is only now reaching a topographic constriction and may
experience a similar destabilization and increase in movement in the near future.
The creeks draining the FDLs modify the permafrost downslope of the lobes, which also may
contribute to accelerated movement.  For example, we observe that the increased sediment
load causes the creeks to jump out of their established channels, causing thermokarsting in the
adjacent ice-rich soils (Gooseff et al., 2009).  Often these creeks disappear within the
permafrost, reappearing farther downhill as springs.  This channel migration lowers the
permafrost table, and increases ground temperature and pore water pressure, facilitating the
movement of the lobe as it slides across the modified terrain.  It is through their drainages that
even the most-distant FDLs are impacting the infrastructure.  The increased sediment
mobilization from FDL movement and destabilization fills ditches and culverts, resulting in





an overtopping hazard to the Dalton Highway and increased maintenance costs. Even FDL-7,
which is across the Dietrich River from the Dalton Highway and TAPS (Figure 1), may affect
the infrastructure. The alluvial fan that is building in front of the lobe has the potential to shift
the Dietrich River channel to the east impinging on the TAPS alignment.
As of August 2015, the eight investigated FDLs range from about 1500 m (FDL-4) to less
than 40 m (FDL-A) from the Dalton Highway (Table S3). Given the rate trends presented in
Figure 10, we can estimate when each FDL will intersect the highway embankment. Based
on its 2015 distance of 39.2m, rate of 5.2m $yr^{-1}$, and correlation coefficient for rate of
movement, we predict that FDL-A will intersect the existing Dalton Highway alignment by
2023. This estimate, however, is based on data from 1955 to 2014, which may have been a
stable time for FDL-A. The recent enlargement of the upper RTS and the formation of large,
persistent transverse cracks across FDL-A mirrors the pattern of instability demonstrated by
FDL-D. These features may forecast rapid downslope movement for FDL-A.
## 5    Conclusions
We present the results of the first comprehensive study of eight FDLs, which include repeated
surface measurements, rock strength testing, soil engineering index property testing, isotope
analysis of infiltration ice, radiocarbon dating, and analysis of historic images of the AOI.
Analysis of these various data sets provided answers to initial questions:
1) The bedrock forming the majority of the catchments has very low to medium strength
and is heavily fractured. These characteristics indicate that the bedrock responds
quickly and easily to physical weathering processes, and thus contributes to the
formation of FDLs.
2) FDLs consist of platy rocks typical of their catchments, along with organic debris, and
an ice-poor soil matrix typically composed of silty sand with varying amounts of
gravel. Massive ice is present within FDLs as infiltration ice, concentrated within
cracks open to the surface. Increased movement and exposure of ice in RTSs leads to
various stages of destabilization, resulting in morphologic differences among the
lobes.
3) Movement of the FDLs within the AOI has been asynchronous since 1955, with FDL-
11 demonstrating significant movement in the 1970s followed by quiescence, while





FDL-7, -D, and -5 currently demonstrate significant movement and/or increasing signs of destabilization. The radiocarbon dating results provide other evidence of asynchronous movement, indicating that FDL-A began to move out of its catchment over 700 years ago, demonstrating either greater or earlier downslope movement than any of the other FDLs.

4) Since 1955, six of the eight investigated lobes demonstrated an increase in movement rates. The formation of surface features, such as cracks, scarps, and RTSs, suggest that the increased movement rates correlate to general instability.

5) We offer a formation scenario of the FDLs after deglaciation of the area, as well as observations on contributing factors to lobe movement and destabilization.

6) Even at a distance, FDLs are impacting infrastructure through increased sediment mobilization. Based on its current distance and rate of movement, we predict that FDL-A will reach the current Dalton Highway alignment by 2023; however, this estimate does not account for the signs of increasing instability in the upper reaches of FDL-A.

While the results of the research presented here have increased our understanding of the composition, morphology, and movement trends of FDLs, this study is not without limitations. 1) Lack of aerial imagery limited the historic image analysis. Many data sets were unusable due to cloud cover, lighting conditions and shadowing, and damage to the film. Analysis of additional imagery could refine the rate trends, and identify the exact timing of rapid movement for FDL-11, -7, and -D. 2) The organic material sampled from FDL-A provided evidence for its initial downslope movement from the catchment. Similar sample collection should be conducted for the other FDLs to increase the understanding of the history of movement. 3) As in previous studies, the observations presented here indicate that FDL movement is closely tied to air temperature. Unfortunately, long-term temperature data does not exist for the immediate area. Future studies could monopolize on the spruce forests in the area by developing a proxy climate record from tree-ring analysis. 4) From our observations, we suspect that infiltration ice comprises a considerable percentage of FDL volume; however, we cannot estimate this volume based on current data. We recommend the use of geophysical methods combined with additional drilling to determine better estimates of ice. 5) Finally, ongoing measurements of surface movement will provide more refined estimates of rates, allowing field identification of destabilization features.





## Acknowledgements

The authors thank their colleagues L. Wirth and M. Slife for their expertise and support on
this project, and Drs. Shur, Kanevskiy, and Stuefer for their valuable input. This work was
funded by grants from the U.S. Department of Transportation (OASRTRS-14-H-UAF-Project
B), the Alaska Department of Transportation and Public Facilities (T2-12-17), and through
generous support from the Alaska Division of Geological & Geophysical Surveys' Capital
Improvement Project and the Alyeska Pipeline Service Company.

## Disclaimer

The views, opinions, findings, and conclusions reflected in this paper are the responsibility of
the authors only and do not represent the official policy or position of the USDOT/OST-R, or
any state agency or entity.

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



Table 1. Summary of available imagery used for historic analysis. USGS is the U.S.
Geological Survey, AHAP stands for Alaska High-Altitude Photography, and DGGS is the
Alaska Division of Geological & Geophysical Surveys. If all FDLs are covered by a given
data set, "NONE" is stated under Limitations.

| Year | Source | Resolution (m) | Limitations in FDL coverage |
|---|---|---|---|
| 1955 | USGS (Aerial) | 1.78 | NONE |
| 1970 | AHAP (Aerial) | 2.0 | NONE |
| 1978 | AHAP (Aerial) | 1.5 | no FDL-5, -4 |
| 1979 | AHAP (Aerial) | 1.5 | only FDL-11, -7, -B |
| 1981 | AHAP (Aerial) | 1.5 | only FDL-D, -5, -4 |
| 1993 | Quantum Spatial (Aerial) | 0.3 | NONE |
| 2007 | DigitalGlobe Ikonos (Satellite) | 1.5 | only FDL-7, FDL-B |
| 2009 | DigitalGlobe WorldView (Satellite) | 0.5 | no FDL-11 |
| 2011 | DGGS (LiDAR) | 1.0 | NONE |
| 2014 | DigitalGlobe WorldView (Satellite) | 0.5 | NONE |





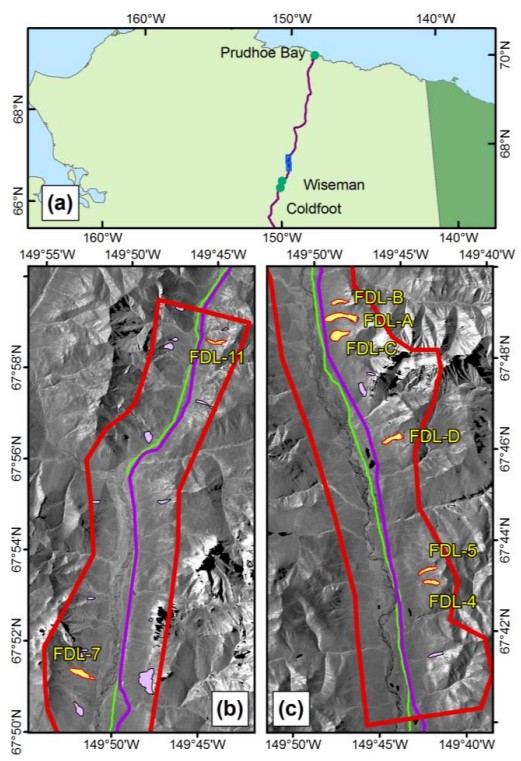

Figure 1. Map of the study area. (a) Location relative to communities along the Dalton
Highway (shown in purple); blue rectangular insets show locations of Area of Interest (AOI).
The northern and southern portions of the AOI (in red) are shown in (b) and (c), respectively.
The eight investigated FDLs are shown in yellow and labeled; other FDLs within the AOI are
shown in lavender. The TAPS alignment is indicated in green; within the AOI the
infrastructure parallels the Dietrich River. (Base map data from ASGDC (2014) and GINA

8 (2001))



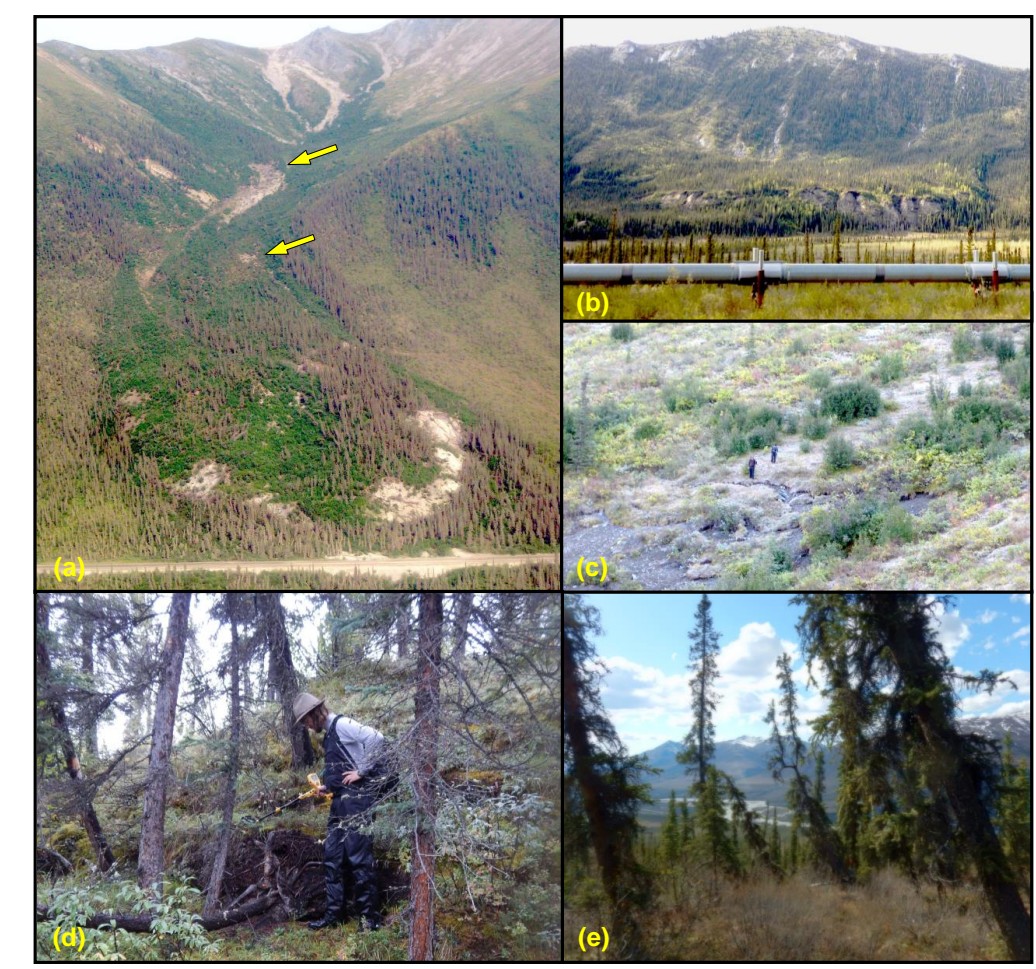

Figure 2. Typical FDL appearance, lobe, and catchment features. (a) FDL-A, originating from a cirque-like catchment; the Dalton Highway is in the foreground (photograph taken in June 2013). Arrows indicate locations of two retrogressive thaw slumps (RTSs). (b) FDL at the base of a slope outside of the AOI that may have formed from a paleo-landslide deposit. The Trans Alaska Pipeline is in the foreground. (c) FDL-11 catchment, showing typical vegetation and recent scarp; two people stand above the scarp for scale. (d) Riser of smaller surface lobe on FDL-C. (e) Trees near the right flank lean progressively towards the center of FDL-5.



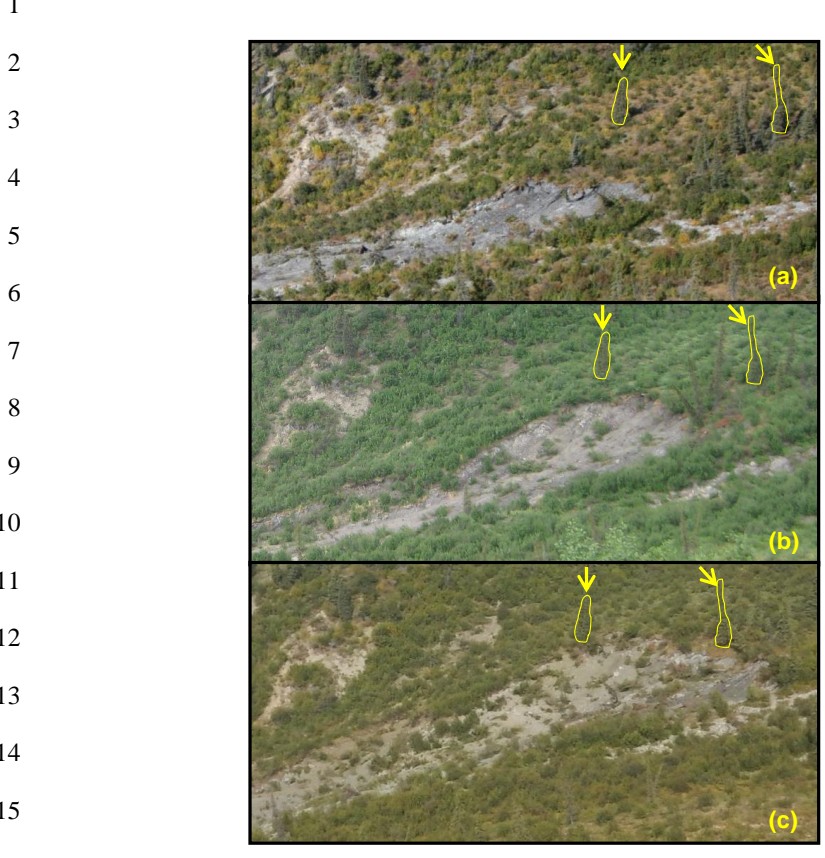

Figure 3. Retrogressive thaw slump (RTS) development on FDL-A: (a) August 2008; (b) June 2013; (c) August 2015. Arrows and outlines indicate the same two trees in all three images for comparison.



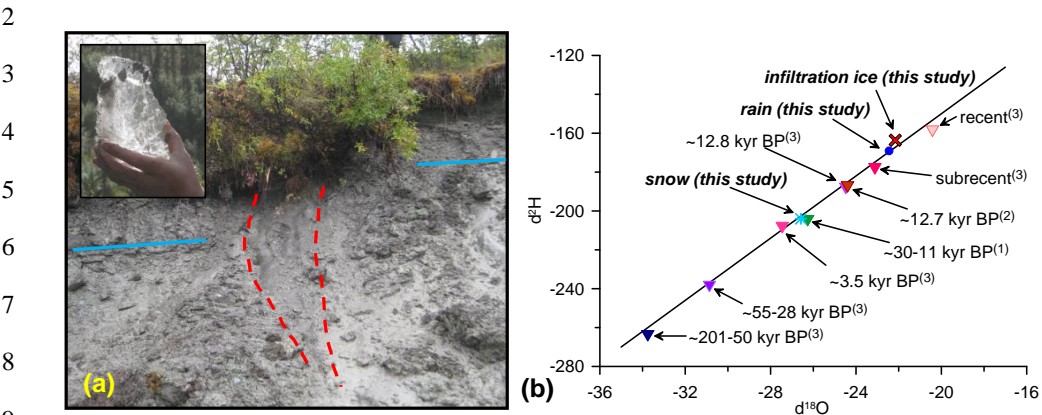

Figure 4. Infiltration ice in FDL-A. (a) Massive ice (outlined by red dashed lines) exposed in
RTS along the left flank in August 2014 (see Figure 2a for location). Offset buried organic
layers are indicated by solid blue lines; inset shows example of clear infiltration ice. (b)
Isotope analysis results; the GMWL is plotted for comparison. Upside-down triangle symbols
represent wedge ice sample values; values taken from the literature are from Douglas et al.
(2011)[1], Meyer et al. (2010)[2], and Meyer et al. (2002)[3].



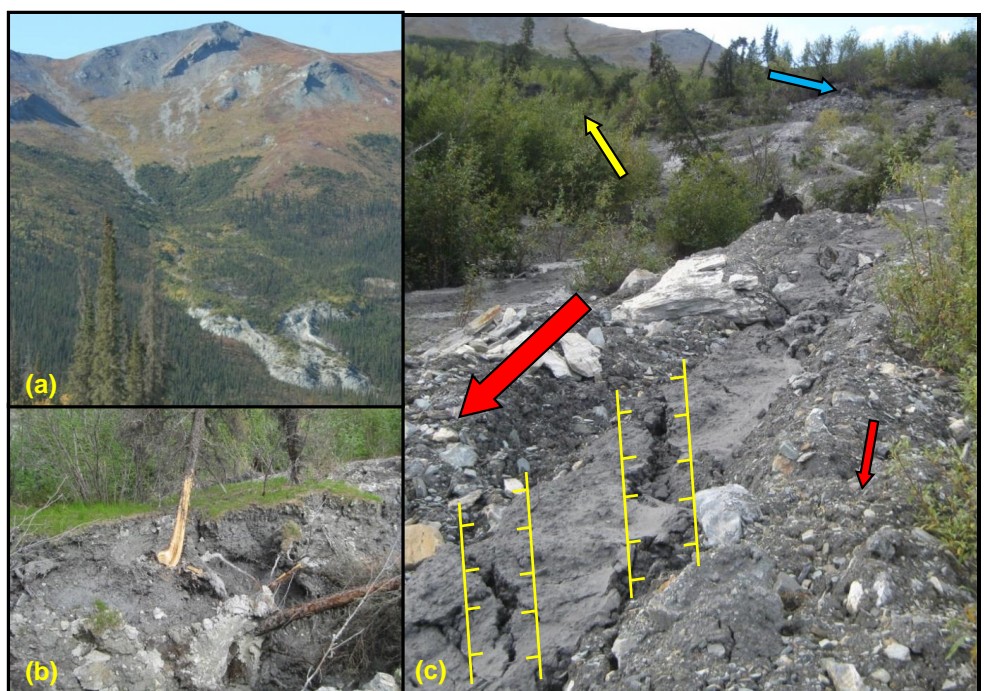

Figure 5.  Features of FDL-7.  (a) Deflation of the main lobe body towards the lower tongue,
and the major RTSs along the right and left flanks.  (b) Vegetation typical of the lower
tongue, including a completely split spruce tree demonstrating about 20 cm of previous
sedimentation along its trunk.  (c) Along the left flank of the lower tongue of FDL-7, differing
rates of movement are indicated by the larger red arrow on the lobe and the smaller red arrow
on the levee (far right).  Echelon cracks are annotated to show extension.  The RTS that was
the source of the debris flow is indicated by the blue arrow, and an area of leaning trees is
indicated by the yellow arrow.



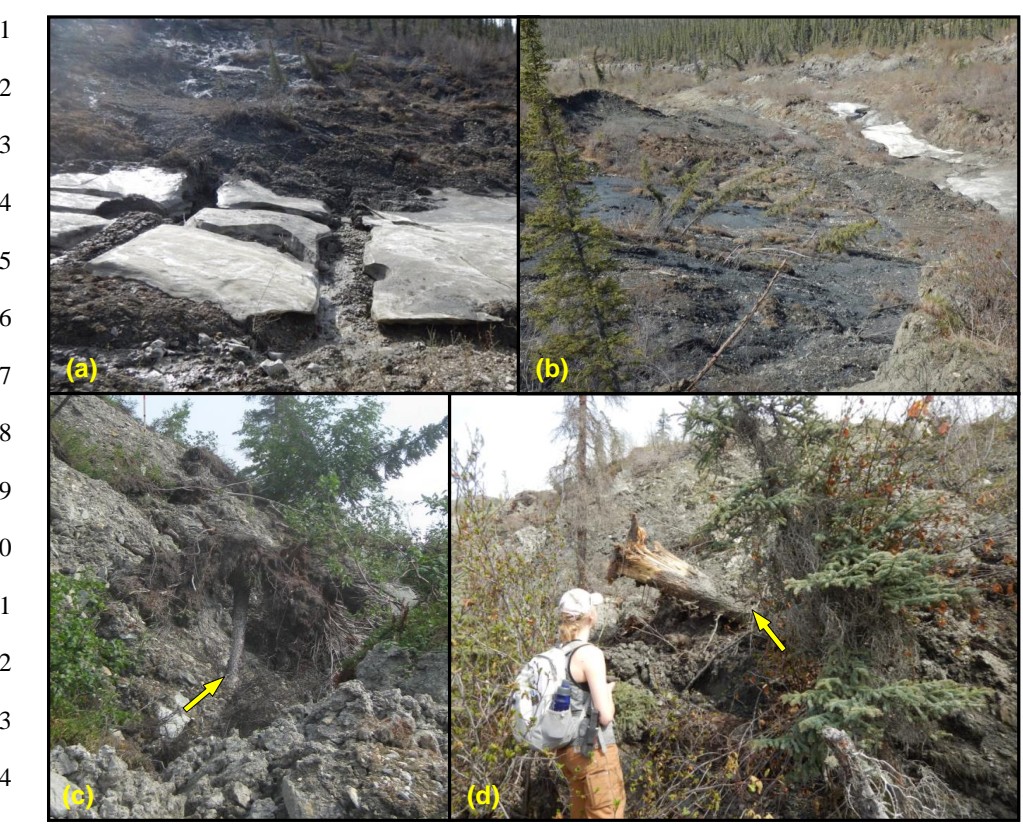

Figure 6.  Destabilization of FDL-D.  (a)  Evidence of ongoing movement throughout the 2014-15 winter and spring, as transverse cracks separate an aufeis deposit on the upper lobe. (b) View down the lobe from the head scarp of one of many RTSs in the catchment, looking over a debris flow originating from the exposed massive ice.  (c) Tree completely upside down with root mass sticking out of a crack. (d) Trunk of tree sticking out of debris at toe of FDL-D.  Yellow arrows in (c) and (d) point to the tree trunks exiting the lobe surface.





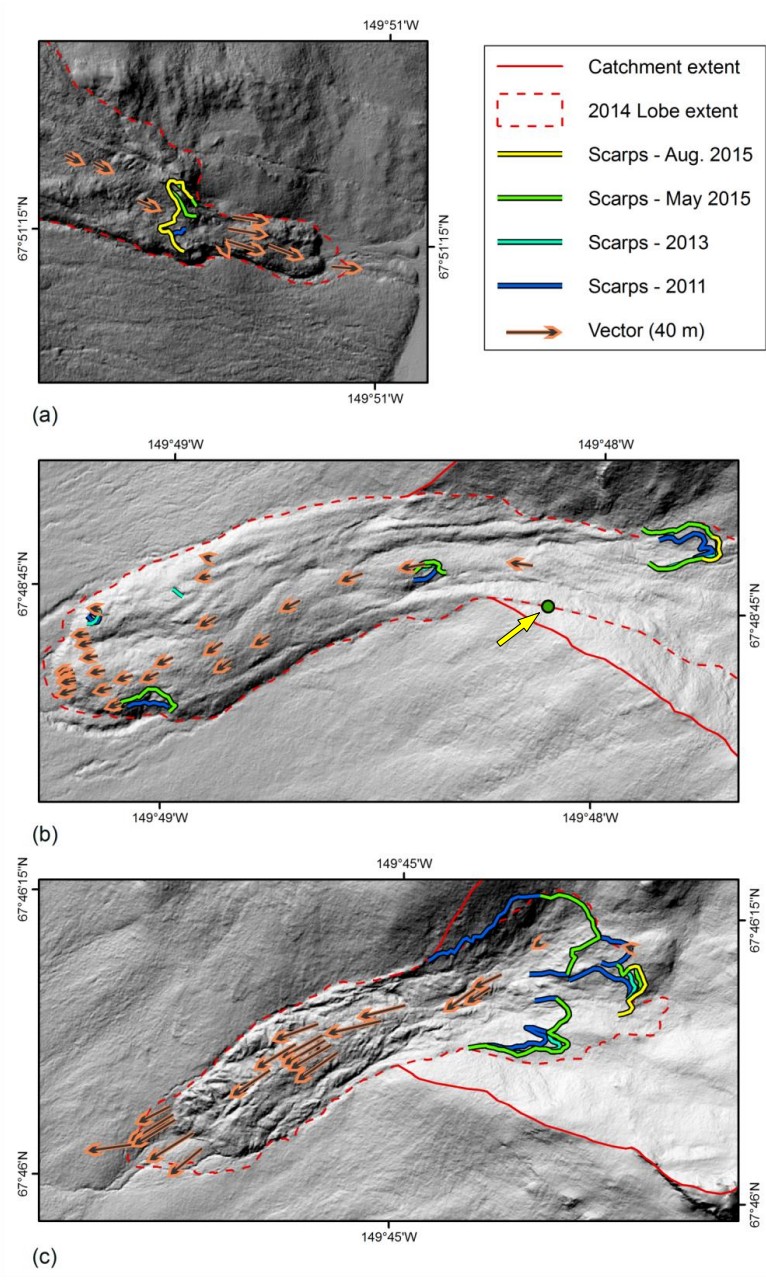

Figure 7.  Vector maps of (a) FDL-7, (b) FDL-A, and (c) FDL–D summarizing movement
measured from June 2013 to August 2015, and RTS development.  The scale of each image is
1:10,000.  Vectors are scaled from the 40 m scale included in the legend.  The arrow and the



1    green dot in (b) indicates the location of sampled organics for radiocarbon dating. (Base maps

2    from 2011 LiDAR data (Hubbard et al., 2011))



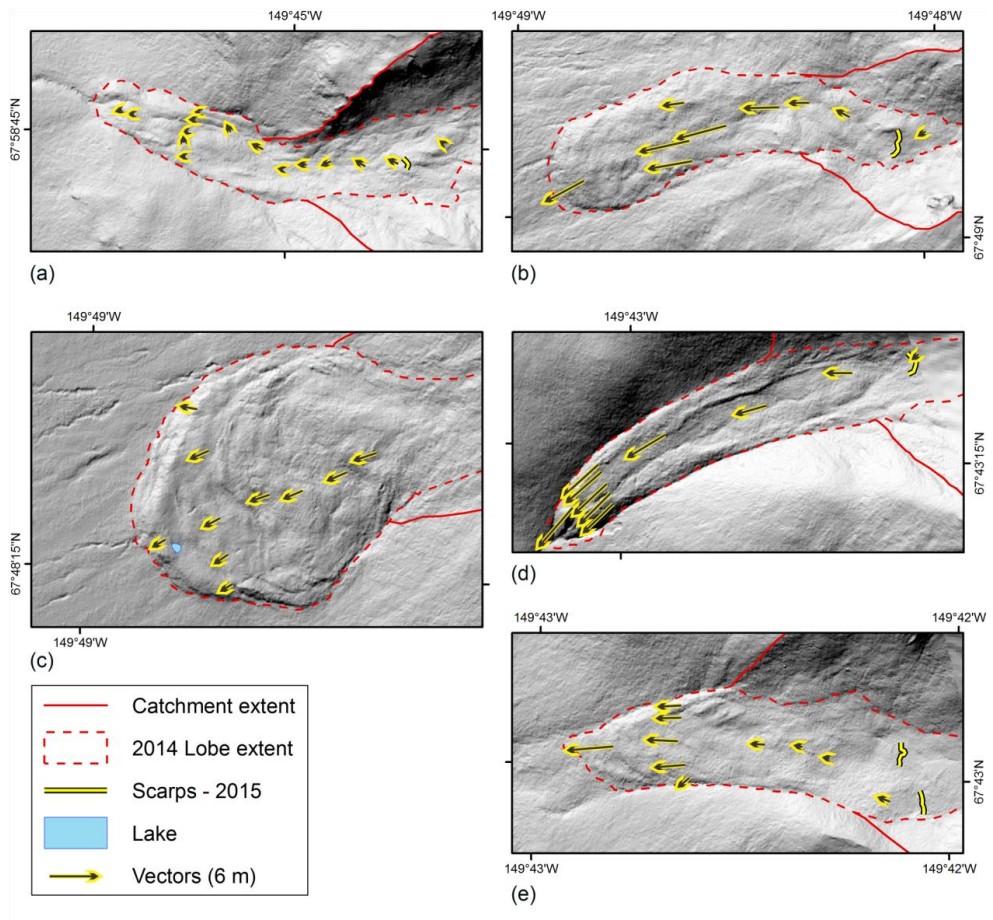

Figure 8. Vector maps of (a) FDL-11, (b) FDL-B, (c) FDL-C, (d) FDL-5, and (e) FDL-4
summarizing movement measured from June 2013 to August 2015, and scarp locations. The
scale of each image is 1:10,000. Vectors are scaled from the 6 m scale included in the legend.
(Base map data from Hubbard et al. (2011) and GINA (2001))





Figure 9. Change in FDL extent from 1955 to 2014: (a) FDL-11, (b) FDL-7, (c) FDL-B, (d) FDL-A, (e) FDL-C, (f) FDL-D, (g), FDL-5, (h) FDL-4. (Base maps from 2011 LiDAR data (Hubbard et al., 2011))





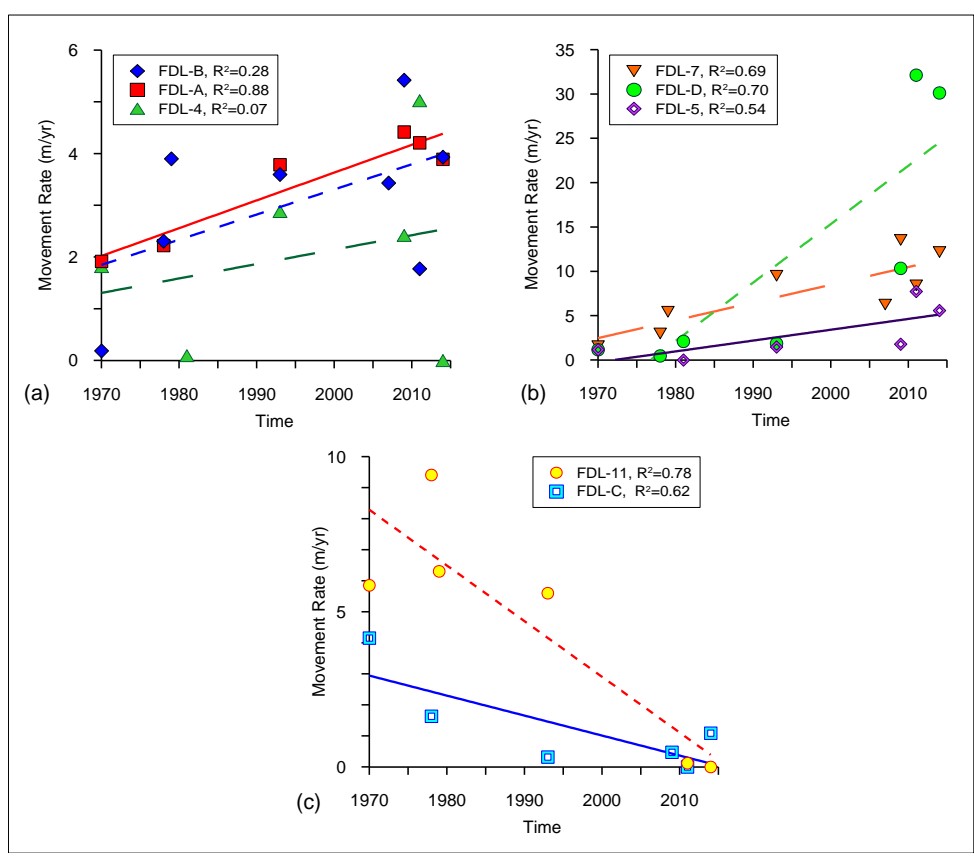

Figure 10. Historic FDL movement rates from 1955 to 2014 for lobes with (a) steadily increasing rates, (b) rapidly increasing rates, and (c) decreasing rates. The coefficient of correlations ($R^2$) for linear trend lines fit to each lobe data set are presented in the figure legends.





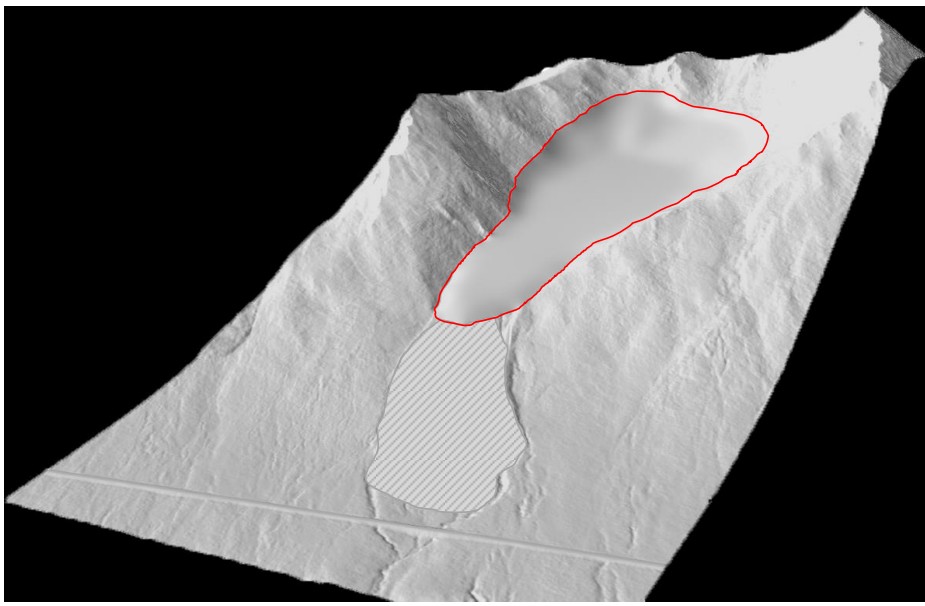

Figure 11. Reconstruction of paleosurface of FDL-A based on bench elevations in its
catchment. The reconstructed lobe is outlined in solid red for visibility. The current lobe
extent that was removed for the reconstruction is indicated by area with gray diagonal lines.
(Base map data from Hubbard et al. (2011))