# Peer review of "overview of eight dynamic features, southern Brooks"

_The Cryosphere, 2016_

## Referee Comment (RC1) · W. Haeberli (Referee) · 28 Feb 2016

General:

The authors present interesting materials and reflections about slope instability and creep phenomena in warm, ice-rich subarctic permafrost. The text is well written and has a logical structure. The illustrations are informative and the conclusions are clear and essential. The findings are innovative and significant as they document an end-member of permafrost-related landform evolution on steep slopes, which is still under-researched: the cumulative flow and deformation of deeply frozen fine-grained/organic materials in forested mountain permafrost. Similarities and contrasts with respect to other members of the envisaged landform evolution – the much better documented

rock glaciers – are correctly mentioned but not really discussed. Here is the main possibility for further improving an already good and welcome contribution. The publications mentioned in the comments below and contained in the subsequent reference list provide important information.

An example is the recommendation by the authors at the end of the Conclusions to use geophysics and drilling in the future. This argument would become stronger if the corresponding possibilities were at least mentioned with reference to the rich experience in rock glacier research. Concerning drilling, for instance, Krainer et al. (2014) show with radiocarbon dating and measurements of borehole temperature and borehole deformation that creep of ice-rich frozen materials and rock glacier formation in warm permafrost in the Alps had taken place since the beginning of the Holocene. Geoelectrical resistivity soundings can provide some indication on the origin and characteristics of subsurface ice (Haeberli and Vonder Mühll, 1996) and in combination with seismic refraction even about ice and unfrozen water contents (Hauck, 2013).

Details:

Abstract, line 18: Information on rock strength is certainly interesting. In the present case more important, however, would be information on the strength and creep properties of the moving material. A recent review on such questions is provided by Arenson et al. (2014).

Abstract, line 26: True but acceleration seems to be predominant – a phenomenon which parallels the recent trend to increasing flow speeds observed on Alpine rock glaciers (see discussion in Deline et al., 2014). A completely synchronous development is hardly to be expected as thermal conditions are not the only factor influencing flow velocities.

Page 2, line 7: Better use "global warming", "atmospheric temperature rise" or so instead of "warming climate". The term "climate" is defined as a statistical average of meteorological conditions and as such cannot "warm" (the expression is popular but

not really scientifically correct).

Page 2, lines 12-15: A more recent and excellent overview is given by Deline et al. (2014).

Page 2, line 30 to Page 3, line 1: The introduction of the term "frozen debris lobes" is an interesting step, especially as the term "rock glacier" has always been questionable (the corresponding phenomenon is neither a rock nor a glacier). In fact, the suggested new name could also be appropriate for what is usually called "rock glacier". The aspect of movement should, however, also be expressed in the new nomenclature.

Page 3, lines 5 – 22: The high subsurface ice content enabling steady-state creep deformation should also be mentioned (cf. core drilling by Krainer et al., 2014 and discussion by Arenson et al., 2014). The term "glacier-cored" should be reconsidered carefully. It relates to a long-outdated geomorphogenetic speculation, which is hardly supported by adequate field measurements (geophysics, core drilling). Of course, buried massive ice can be preserved within permafrost. For simple size reasons, however, such buried ice is in most cases remains from ice patches, avalanche deposits or glacierets rather than real "glaciers". Making a full stop after "Pleistocene glaciation" would help avoiding such discussions.

Page 4, lines 16-18: Where are the temperature data from? What depths and times do they cover? In which (lower, upper) parts of which FDL were they taken? And to what sites in the adjacent permafrost were they compared? This important information should be precise.

Page 8, line 32: Should the high frost susceptibility of such silty sand be mentioned here? It could be a key factor concerning subsurface ice content and creep mode.

Page 9, lines 6-7: Can more information be given on this drilling? How representative is the information on extremely small ice contents? How was this ice content determined? Was melting of core-ice during drilling prevented by cold-air cooling or so?

**TCD**

Were temperatures at depth measured here? Where was the exact position of this drilling?

Page 10, line 16: This occurrence of massive ice and ice-rich soil here seems to be in strong contrast with the extremely low ice contents found in the drilling on FDL-A (cf. previous remark). Is this contrast real and, if yes, can it be explained?

Page 12, lines 17-27: Reference should be made to the dated permafrost core through an active rock glacier described by Krainer et al. (2014), which documents a similar evolution for rock glaciers in the Alps. The authors should also have a look at the concepts developed on the basis of core drilling and borehole measurements already in the late 1990s for rock glacier evolution over time (Haeberli et al., 1998). These concepts are comparable to the ideas presented here but provide more detail about flow physics and the internal layering of the creeping body. They especially also consider the phenomenon that material from the more rapidly moving surface falls down over the steep front and is then overridden by the more slowly advancing lower parts of the front.

Page 13, line 16: Again – compare with rock glacier datings (Krainer et al., 2014 and other references provided there).

Page 13, lines 28-30: The possibilities of geophysical soundings could be mentioned here and primary results from such measurements on rock glaciers could be summarised.

Page 14, lines 6-7: Why are the debris flows increasing the surface temperature? Provide a brief explanation of the physical process involved.

Page 14, lines 10-12: Ikeda et al. (2008) document and discuss detailed field evidence on this process chain from drilling and borehole measurements.

Page 15, line 16: Better write " . . . study of eight FDLs near the Dalton Highway in the Brooks Range, which . . ." for the readers who primarily look at the conclusions.

[Figure]

Page 16, line 25: In view of the still strongly limited temperature data and the evolution in time, it could be more appropriate to write: " . . . movement changes which may be tied to changes in air temperature." (The movement itself is not tied to air temperature in a straightforward way but rather via a complex process chain).

Page 16, lines 29-30: Concerning geophysical soundings and drilling refer to the general comments at the beginning of this review.

Caption of Figure 1: Is there only one blue rectangular inset in (a)?

Caption of Figure 5: What exactly is meant with the term "Deflation"? This term usually stands for erosion by wind. Is this meant here?

References:

Arenson, L., Colgan, W. and Marshall, H.P. (2014): Physical, thermal and mechanical properties of snow, ice and permafrost. In: Haeberli, W., Whiteman, C. (Eds.), Snow and Ice-related Hazards, Risks and Disasters. Elsevier, 35-75. Deline, P., Gruber, S., Delaloye, R., Fischer, L., Geertsema, M., Giardino, M., Hasler, A., Kirkbride, M., Krautblatter, M., Magnin, F., McColl, S., Ravanel, L. and Schoeneich, P. (2014). Ice loss and slope stability in high-mountain regions. In: Haeberli, W., Whiteman, C. (Eds.), Snow and Ice-related Hazards, Risks and Disasters. Elsevier, 303-344. Haeberli, W. and Vonder Mühll, D. (1996): On the characteristics and possible origins of ice in rock glacier permafrost. Zeitschrift für Geomorphologie N.F., 104, 43-57. Haeberli, W., Hoelzle, M. Kääb, A., Keller, F., Vonder Mühll, D. and Wagner, S. (1998): Ten years after drilling through the permafrost of the active rock glacier Murtèl, eastern Swiss Alps: answered questions and new perspectives. Proceedings of the Seventh International Conference on Permafrost, Yellowknife, Canada, Collection Nordicana, 57, 403-410. Hauck, C. (2013): New concepts in geophysical surveying and data interpretation for permafrost terrain. Permafrost and Periglacial Processes 24, 131-137. doi:10.1002/ppp.1774 Ikeda, A., Matsuoka, N., and Kääb, A. (2008): Fast deformation of perennially frozen debris in a warm rock glacier in the Swiss Alps: an effect of liquid

water. Journal of Geophysical Research 113, F01021. doi.org/10.1029/2007JF000859
Krainer, K., Bressan, D., Dietre, B., Haas, J.N., Hajdas, I., Lang, K.,Mair, V., Nickus, U., Reidl, D., Thies, H. and Tonidandel, D. (2014): A 10,300-year-old permafrost core from the active rock glacier Lazaun, southern Ötztal Alps (South Tyrol, northern Italy). Quaternary Research 83 (2), 324–335. doi:10.1016/j.yqres.2014.12.005

---

## Referee Comment (RC2) · Anonymous Referee #2 · 11 Mar 2016

The paper addresses distribution and dynamics of several frozen debris lobes in the Brooks Range. The manuscript provides much information about these features, which not only are clear signs for permafrost, but also pose a possible threat to infrastructure in the region when accelerating or decaying. Thus, this manuscript is of high interest for the general cryosphere community and should be published after a revision.

When writing this review, I have also read the review provided by W. Haeberli (Reviewer 1), and most of his comments I agree and will not duplicate here. From my point of view, the following major issues arise:

1. Writing style: The paper is long and wordy, the style reminds me of an oral lecture (much "we have . . ." etc.), including many details which are important in a report to e.g.

a government agency, but not in a comprehensive scientific publication. The paper contains some redundant information, like "rain has exposed ice" is mentioned some times. The paper could be re-structured and shortened.

2. Introduction: Is very long, ranging from an historic overview about the research development of the slope features to a mini review about the term "rock glacier". I would suggest shortening this and stick to scientific important points.

3. Setting: A "Setting" chapter is missing as far as I can see. You use the "Introduction" partly to describe the setting, however, I think readers not familiar with the region would like to know a bit more about the geophysiographic conditions including key values of earlier investigations as given in p. 4, l. 4 ff.

4. Methods: 2.1. is very wordy and could be shortened. In the result/discussion you introduce new methods, like dating organic layers (p. 13) or the collection of creek samples (p. 9) etc. This should be introduced in the method section, and subsequently described in the Result chapter.

5. Results: The results chapter is much longer than the Discussion chapter, often because you already give interpretations of observations here, which would be good for a discussion. As mentioned above, also new methods are introduced here.

6. Discussion: The discussion is poor. It contains paragraphs which would be good in a "Setting" chapter which includes previous investigations (e.g. p. 12) or results (p. 13, organic layer), but is lacking a scientific discussion such as a comparison and relevance to other studies, rock glaciers etc. This is also pointed out by reviewer 1, and highlighted nicely in that review. Maybe it is better to move your little rock glacier review from the Introduction to a discussion chapter, and really discuss your findings with the literature focusing on debris bodies containing more coarse material.

7. Your timing based on the one radio-carbon date. You must be very careful here, normally I would say that you cannot say anything based on one dating from one site.

You can mention the date, but one dating does not justify a strong conclusion. But its is good to discuss against other studies, as suggested by reviewer 1.

8. Conclusions: The bullet point conclusions sound ok and are mostly justified by your observations and measurements (beside the dating). The second part is again a Discussion and not a conclusion, so you should remove it or move it to an appropriate place in the discussion.

Some details: p. 3, l. 21: You may use the term "moraine-derived". L 27 ff: this all is a discussion p. 4, 2nd para: This is typical info for a "Setting" chapter p. 5, l1: This whole paragraph can be removed p. 7, l 16-19: remove p. 8, l 21: "The rounder . . .": This is interpretation and should be addressed in an discussion l. 30: agreeing with reviewer 1: also what commonly is termed "rock glaciers" are of course frozen and moving debris lobes, only consisting of coarser material. And maybe "frozen debris lobe" is actually a better term than "rock glacier", which you should discuss in the end. p. 9, 2nd para: Again a lot of interpretations of the observations which sound reasonable, but should come into a discussion. l. 20: This is a figure text, avoid explaining figures in detail in the main text. 3rd para: See above, not mentioned in method chapter p. 10: Here you mix observation and velocity measurements, which you present in subsequent paragraphs, it is a bit hard to follow this structure. p. 11, l. 13: Remove this sentence, if the thing would not move downhill I would question your measurements. . . l. 27: Again parts of the paragraph describe a method you used. p. 13, l 1: You mention now "benches", what is this? Maybe explain this in a setting part. In the following you again introduce a new method (dating, see above)

---

## Author Comment (AC1) · 5 Apr 2016

Response to Reviewer 1 (Dr. Haeberli) for manuscript TC-2016-1, "Frozen debris lobe morphology and movement: an overview of eight dynamic features, southern Brooks Range, Alaska"

Dr. Haeberli, first of all, thank you very much for your kind overview and constructive comments. Many of them serve as a teaching tool, bringing some relevant references to the authors' attention. In what follows, your initial comment is presented, followed by the authors' response.

Abstract, line 18: Information on rock strength is certainly interesting. In the present

case, more important, however, would be information on the strength and creep properties of the moving material. A recent review on such questions is provided by Arenson et al (2014). Many of the same authors from this paper presented information on the strength properties of the soil from FDL-A from frozen direct shear tests in Simpson et al. This article has been accepted but is not yet published; a PDF of the draft is available at this link: http://eeg.geoscienceworld.org/content/early/2015/11/04/EEG-1728.full.pdf+html?sid=0019d85f-9d60-4481-a3d9-21bc44167e04. We did not include the strength properties here so as not to repeat the previous paper. We have not run creep tests on the material.

Abstract, line 26: True but acceleration seems to be predominant – a phenomenon which parallels the recent trend to increasing flow speeds observed on Alpine rock glaciers (see discussion in Deline et al 2014). A completely synchronous development is hardly to be expected as thermal conditions are not the only factor influencing flow velocities. The text has been revised to reflect the spirit of this comment: "Analysis of historic imagery indicates that movement of the eight investigated FDLs has been asynchronous over the study period, and since 1955, there is an overall increase in movement rates of the investigated FDLs."

Page 2, Line 7: Better use "global warming", "atmospheric temperature rise" or so instead of "warming climate". The term "climate" is defined as a statistical average of meteorological conditions and as such cannot "warm" (the expression is popular but not really scientifically correct). Revised as suggested.

Page 2, lines 12-15: A more recent and excellent overview is given by Deline et al 2014. Revised the text to include related material from reference as follows: "Warmer temperatures lead to deeper active layer depths resulting in increased water infiltration; ice within the soil or debris melts, causing loss of soil strength, accelerated movement, and potential debris flows or total collapse (Deline et al., 2015, Geertsema et al. . .”

Page 2, line 30 to Page 3, line 1: The introduction of the term "frozen debris lobes" is an

interesting step, especially as the term "rock glacier" has always been questionable (the corresponding phenomenon is neither a rock nor a glacier). In fact, the suggested new name could also be appropriate for what is usually called "rock glacier". The aspect of movement should, however, also be expressed in the new nomenclature. This new term came about through the review of the paper Daanen et al. (2012) (I believe you were involved in that review). I spoke with Ronald Daanen about how to include movement into the term. These features move mostly by shear, with secondary or minor internal flow/creep. What would the reviewer suggest here? We are concerned about including too much process into the term.

Page 3, lines 5-22: The high subsurface ice content enabling steady-state creep deformation should also be mentioned (cf core drilling by Krainer et al 2014 and discussion by Arenson et al 2014). The term "glacier-cored" should be reconsidered carefully. It relates to a long-outdated geomorphogenetic speculation, which is hardly supported by adequate field measurements (geophysics, core drilling). Of course, buried massive ice can be preserved within permafrost. For simple size reasons, however, such buried ice is in most cases remains from ice patches, avalanche deposits or glacierets rather than real "glaciers". Making a full stop after "Pleistocene glaciation" would help avoiding such discussions. This text was moved to the Discussion, and the last reference was revised as suggested. The remaining text was heavily revised, including a comparison to the data presented by Krainer et al. and a summary of movement through creep.

Page 4, lines 16-18: Where are the temperature data from? What depths and times do they cover? In which (lower, upper) parts of which FDL were they taken? And to what sites in the adjacent permafrost were they compared? This important information should be precise. This data is included in the Simpson et al. paper, and so again, we avoid repeating that content here. We have updated Figures 2a and 7b (now renumbered to Figure 5b) to show the location of the 2012 borehole in FDL-A in which the temperature measurements are taken. We also revised the text to describe more
about the measurements and when they were obtained (which changed the values initially provided): "The significant movement within the shear zone severed the instrumentation approximately one month after its installation; however, we are still able to collect subsurface temperature and movement measurements from the upper 20.6m of the M-IPI. Temperatures measured from 15 to 20.6m from January 2014 through August 2015 were stable at -0.85°C, whereas the temperature of the adjacent permafrost at 3m from the same time period was -2.1°C."

Page 8, line 32: Should the high frost susceptibility of such silty sand be mentioned here? It could be a key factor concerning subsurface ice content and creep mode. Based on field observations, I do believe that the debris lobe soil is frost susceptible; however, we have not yet performed frost heave tests on the soil. I also believe that the soil has a significant unfrozen water content, but this is also speculation as we have not yet tested it (there are plans to do this in the near future). While frost heave may contribute to the creep/flow component, this mode of movement is secondary to the tremendous shear that these features experience. Added the following lines to illustrate this point within the new Study Site and Background section: "Sub-surface measurements within FDL-A indicate that this frozen debris lobe moves predominantly through shear in a zone 20.6 to 22.8m below ground surface (bgs), with temperature-dependent internal flow as a secondary movement mechanism (Darrow et al., 2015; Simpson et al., in press). For example, between September 2012 and August 2015, FDL-A moved 13.8m through shear and only 1.9m through internal flow, for a total displacement of 15.7m at the main borehole location."

Page 9, lines 6-7: Can more information be given on this drilling? How representative is the information on extremely small ice contents? How was this ice content determined? Was melting of core-ice during drilling prevented by cold-air cooling or so? Were temperatures at depth measured here? Where was the exact position of this drilling? As previously mentioned, the drilling program and results obtained are described in greater detail in Simpson et al. (in press). We are concerned about repeating too much of that content in this paper. Revised the text regarding excess ice content as follows: "Boreholes from the 2012 subsurface investigation intercepted no massive ice, and all samples obtained from the drilling were ice-poor (i.e., samples contained no excess ice and volumetric moisture contents (averaging 31%) were less than the calculated porosity of the soil)."

Page 10, line 16: This occurrence of massive ice and ice-rich soil here seems to be in strong contrast with the extremely low ice contents found in the drilling on FDL-A. Is this contrast real, and if yes, can it be explained? Yes, it is real! This also was a surprise to us. The soil within the lobe is indeed ice-poor, at least in our boreholes. We try to explain that the massive ice forms in the cracks open at the surface (i.e., infiltration ice), which then become covered and thermally protected until re-exposed.

Page 12, lines 17-27: References should be made to the dated permafrost core through an active rock glacier described by Krainer et al 2014, which documents a similar evolution for rock glaciers in the Alps. The authors should also have a look at the concepts developed on the basis of core drilling and borehole measurements already in the late 1990s for rock glacier evolution over time (Haeberli et al., 1998). These concepts are comparable to the ideas presented here but provide more detail about flow physics and the internal layering of the creeping body. They especially also consider the phenomenon that material from the more rapidly moving surface falls down over the steep front and is then overridden by the more slowly advancing lower parts of the front. This text has been significantly revised, to include the suggested references.

Page 13, line 16: Again, compare with rock glacier datings (Krainer et al., 2014 and other references provided). After reviewing the Krainer et al. (2015) paper, we are looking at two different things. That reference discusses the continual accumulation of the rock glacier and periods of permafrost instability in the past. It indicates a continuous stratigraphy within a small feature. What we are trying to communicate with this date is a time when FDL-A may have started to move out of its catchment and move quickly downslope. This is after its growth period within the catchment (similar to the reference). In fact, it would be difficult to reconstruct a similar chronology, since the lobe geometry is dramatically different now from when it was smaller within the catchment.

Page 13, lines 28-30: The possibilities of geophysical soundings could be mentioned here and primary results from such measurements on rock glaciers could be summarized. We have experimented with a few geophysical techniques already. Seismic refraction did not get us deep enough with the available equipment to discern the shear zone, and we were not able to drill where the seismic lines were located, thus not ground-truthing the results. We also tried the passive seismic method, but this method was unsuccessful since the subsurface layers are either thin or have similar seismic properties. We do intend to try an induced electromagnetic method this year. We anticipate that Induced Polarization Tomography (IPT) will be most successful to penetrate to the depth of the shear zone, and to locate water within the lobe; however, its employment depends on funding. The text was revised to include the references on geophysics.

Page 14, lines 6-7: Why are debris flows increasing the surface temperature? Provide a brief explanation of the physical process involved. This text was revised to: "The meltwater forms debris flows that cover a larger area of the lobe, changing the moss-covered surface to bare mineral soil, which increases the surface temperature and repeats the cycle. . ." The debris flows change the surface thermal regime, eventually causing a shift in the vegetation.

Page 14, lines 10-12: Ikeda et al. (2008) document and discuss detailed field evidence on this process chain from drilling and borehole measurements. Added the following text to a different but also relevant portion of the Discussion: "Ikeda et al. (2008) document a similar process in a rock glacier in the Swiss Alps. In the rock glacier, movement formed tensile cracks, allowing snow melt to penetrate into voids, decreasing effective stress and increasing movement rates."

Page 15, line 16: Better write ". . .study of eight FDLs near the Dalton Highway in the

Brooks Range, which..."for readers who primarily look at the conclusions. Revised as suggested.

Page 16, line 25: In view of the still strongly limited temperature data and the evolution in time, it could be more appropriate to write: "...movement changes which may be tied to changes in air temperature." (The movement itself is not tied to air temperature in a straightforward way but rather via a complex process chain). Revised as suggested.

Page 16, lines 29-30: Concerning geophysical soundings and drilling refer to the general comments at the beginning of this review. See response above to related comment. This text has moved into the Discussion section. Added suggested method to this sentence.

Caption of Figure 1: Is there only one blue rectangular inset in (a)? Yes, the rectangles are small, but there are two (one to the north for the (b) frame, and one to the south for the (c) frame). They tend to merge together because of the scale. If you increase the size of the image, you can see two.

Caption of Figure 5: What exactly is meant with the term "deflation"? This term usually stands for erosion by wind. Is this meant here? Here we did not imply erosion by wind, but rather how a hot air balloon may deflate. We tried to describe the center portion reduces in thickness and flows out the middle (like a tube of toothpaste) while the sides remain relatively intact. Is there a more appropriate geomorphic process term that should be used to describe this?

---

## Author Comment (AC2) · 5 Apr 2016

Response to Reviewer 2 for manuscript TC-2016-1, "Frozen debris lobe morphology and movement: an overview of eight dynamic features, southern Brooks Range, Alaska"

We thank Reviewer 2 for the thorough review of our paper. It has been difficult to structure this paper in a way that makes sense, and many of the reviewer's comments will help to address this issue. In the text below, your initial comment is presented followed by the authors' response.

1. Writing style: The paper is long and wordy, the style reminds me of an oral lecture

(much "we have..." etc.), including many details which are important in a report to e.g. a government agency, but not in a comprehensive scientific publication. The paper contains some redundant information, like "rain has exposed ice" is mentioned some times. The paper could be re-structured and shortened. This specific comment contradicts Reviewer 1's initial comment "The text is well written and has a logical structure." In last two decades, the lead author has been a push in the professional geotechnical community to use the active voice, rather than the passive voice which is traditional and "what we all learned". This leads to 'we haves', where appropriate. Can the reviewer please indicate which details are superfluous to a scientific publication? A check of the document indicated that the "rainfall" reference occurred twice; the second reference was deleted.

2. Introduction: is very long, ranging from an historic overview about the research development of the slope features to a mini review about the term "rock glacier". I would suggest shortening this and stick to scientific important points. Originally, the Introduction was shorter. The editor who initially reviewed the article instructed the authors to provide a greater literature review. At this point, the literature review of rock glaciers has been moved to the Discussion section.

3. Setting: A "Setting" chapter is missing as far as I can see. You use the "Introduction" partly to describe the setting, however, I think readers not familiar with the region would like to know a bit more about the geophysiographic conditions including key values of earlier investigations as given in p.4, l. 4 ff. Added a Study Site and Background section, providing some general information about the AOI, and moving the discussion of previous FDL work to this section.

4. Methods: 2.1. is very wordy and could be shortened. In the results/discussion you introduce new methods, like dating organic layers (p. 13) or the collection of creek samples (p. 9) etc. This should be introduced in the method section, and subsequently described in the result chapter. At attempt was made to shorten the method descriptions. Moved referenced text into the Methods section (as indicated below).

5. Results: The results chapter is much longer than the Discussion chapter, often because you already give interpretations of observations here, which would be good for a discussion. As mentioned above, also new methods are introduced here. Methods have been moved as suggested; the text was revised with this comment in mind. Also, a large portion of the text describing the destabilization process has been moved to the Discussion.

6. Discussion: The discussion is poor. It contains paragraphs which would be good in a "Setting" chapter which includes previous investigations (e.g. p. 12) or results (p. 13, organic layer), but is lacking a scientific discussion such as a comparison and relevance to other studies, rock glaciers etc. This is also pointed out by reviewer 1, and highlighted nicely in that review. Maybe it is better to move your little rock glacier review from the Introduction to a discussion chapter, and really discuss your finding with the literature focusing on debris bodies containing more coarse material. Moved relevant parts of organic dating discussion to methods and results. Moved rock glacier review into this section, comparing and contrasting to FDLs. The rest of the Discussion has been heavily revised.

7. Your timing based on the one radio-carbon date. You must be very careful here, normally I would say that you cannot say anything based on one dating from one site. You can mention the date, but one dating does not justify a strong conclusion. But it is good to discuss against other studies, as suggested by reviewer 1. We acknowledge the reviewer's concern of one radiocarbon date; however, we do feel that it is useful preliminary information regarding the formation/movement of these features. This text has been significantly revised, including acknowledgment of having only one date.

8. Conclusions: The bullet point conclusions sound ok and are mostly justified by your observations and measurements (beside the dating). The second part is again a Discussion and not a conclusion, so you should remove it or move it to an appropriate place in the Discussion. Indicated text has been moved as suggested.

[Figure]

Specific comments:

p. 3, l. 21: You may use the term "moraine-derived". This text has been rewritten.

L 27 ff: This is all a discussion. The referenced text has been moved and integrated into the Discussion section.

p. 4, 2nd para: This is typical information for a Setting chapter. The referenced text has been moved to a new Study Site and Background section.

p. 5, l 1: This whole paragraph can be removed. Deleted the referenced text.

p. 7, l 16-19: Remove Deleted as suggested.

p. 8, l 21: This is interpretation and should be addressed in discussion. This is a difficult comment to address. While the referenced text does contain some interpretation, it is part of a summary of the field observations (results of the field work). If the referenced text were pulled out to move to the Discussion section, it would necessitate additional text to explain its relevance there, making the paper longer.

L 30: agreeing with reviewer 1 – also what commonly is termed rock glaciers are of course frozen and moving debris lobes, only consisting of coarser material. And maybe frozen debris lobe is actually a better term than rock glacier, which you should discuss in the end. Yes, the term FDL could be broader to include rock glaciers; however, we want to stress that these features we are describing in the Brooks Range are fundamentally different in size, composition, vegetation coverage, and mechanisms of movement than what is now called a rock glacier. We agree that the discussion should occur, but are concerned about increasing the paper length even more.

p. 9, 2nd para: Again, a lot of interpretations of the observations which sound reasonable, but should come into a discussion. Most of this paragraph is observations, with little interpretation; deleted the phrase about rainfall helping to melt ice.

l. 20: This is a figure text, avoid explain in figures in detail in the main text. Revised

to "The exposed massive ice corresponds with an open surface crack, with a buried organic layer vertically offset to its right and left, indicating downslope movement." This does not repeat the figure caption.

3rd para: refers to 3rd paragraph? Says not mentioned in method chapter. We are a little unclear as to what the reviewer is referencing. With certain assumptions in mind, we revised the text to: "Figure 4b is a presentation of the isotope analysis results with the GMWL and isotope values from massive ice bodies taken from the literature, including Pleistocene wedge ice..." We introduced the GMWL in the Methods section.

p. 10: Here you mix observation and velocity measurements, which you present in subsequent paragraphs, it is a bit hard to follow this structure. Moved a large portion of this text to the Discussion section.

p. 11, l. 13: – Remove this sentence, if the thing would not move downhill, I would question your measurements. The point of this sentence was to indicate that the FDL demonstrates minimal spreading from the centerline. It has been revised to: "Movement is generally parallel to each FDL's longitudinal profile."

l. 27: Again parts of the paragraph describe a method you used. The referenced text was revised, with portions moved to the Methods section.

p. 13, l 1: you mention now "benches" what is this? Maybe explain this is a setting part. In the following you again introduce a new method (dating see above). Moved the organic soil sampling and methodology to the Methods section. Moved the result into the Results section. Kept part within the Discussion related to timing.

---

## Author Response (AR2)

Thank you, Editor Isaksen, for your work on reviewing this paper.  I have made the changes that you suggested in the abstract and on page 2.

Sincerely,

Margaret Darrow